# Direct and Indirect Proof of SARS-CoV-2 Infections in Indigenous Wiwa Communities in North-Eastern Colombia—A Cross-Sectional Assessment Providing Preliminary Surveillance Data

**DOI:** 10.3390/vaccines9101120

**Published:** 2021-10-01

**Authors:** Gustavo Concha, Hagen Frickmann, Anke Oey, Monika Strengert, Lothar Kreienbrock, Simone Kann

**Affiliations:** 1Organization Wiwa Yugumaiun Bunkauanarrua Tayrona (OWYBT), Department Health Advocacy, Valledupar 2000001, Colombia; gustavoconcha16@gmail.com; 2Department of Microbiology and Hospital Hygiene, Bundeswehr Hospital Hamburg, 20359 Hamburg, Germany; frickmann@bnitm.de or; 3Institute for Medical Microbiology, Virology and Hygiene, University Medicine Rostock, 18057 Rostock, Germany; 4Institute for Biometry, Epidemiology and Information Processing, University of Veterinary Medicine Hannover, 30559 Hannover, Germany; Anke.Oey@tiho-hannover.de (A.O.); lothar.kreienbrock@tiho-hannover.de (L.K.); 5Helmholtz Centre for Infection Research, 38124 Braunschweig, Germany; Monika.Strengert@helmholtz-hzi.de; 6TWINCORE Centre for Experimental and Clinical Infection Research, 30625 Hannover, Germany; 7Medical Mission Institute, 97074 Würzburg, Germany

**Keywords:** COVID-19, indigenous, Colombia, surveillance, prevalence, incidence, outbreak

## Abstract

To provide initial data on local SARS-CoV-2 epidemiology and spread in indigenous communities in north-eastern Colombia, respiratory swabs and serum samples from volunteers of indigenous communities were examined in March and April 2021. Samples from non-indigenous Colombians from the same villages were included as well. While previous exposure to SARS-CoV-2 was assessed by analysing serum samples for IgG and IgM with a rapid antibody point-of-care-test (POCT), screening for active infections was carried out with an antigen POCT test and real-time PCR from nasal swabs. In 380 indigenous and 72 non-indigenous volunteers, 61 (13.5%) active infections and an additional 113 (25%) previous infections were identified using diagnostic serology and molecular assays. Previous infections were more frequent in non-indigenous volunteers, and relevant associations of clinical features with active or previous SARS-CoV-2 infections were not observed. Symptoms reported were mild to moderate. SARS-CoV-2 was frequent in the assessed Colombian indigenous communities, as 38.5% of the study participants showed signs of exposure to SARS-CoV-2, which confirms the need to include these indigenous communities in screening and vaccination programs.

## 1. Introduction

First reported in Wuhan, China, in December 2019 [1], the global spread of COVID (coronavirus disease)-19-associated SARS-CoV-2 (severe acute respiratory syndrome-coronavirus-2) led to a pandemic with dramatic medical and social consequences. Based on the rapid global transmission of SARS-CoV-2, concerns arose on the safety and health of indigenous communities living in retracted areas and resource-poor conditions where strict hygiene measures are difficult to enforce. As early as April 2020, several medical journals expressed their fear for the health of such particularly endangered individuals [2], stressed the scarcely available surveillance data [3] and called for action to optimally protect indigenous societies from the disruptive effects of the global pandemic [4].

In May 2020, Bolivian, US American and French scientists published an action plan for the protection of indigenous communities in the Bolivian Amazon by voluntary isolation. It also included educational, outreach and preparation strategies for a first phase of containment, patient management and quarantine for a second phase [5]. The topic remained, however, controversial [6,7,8,9,10,11], with some authors even considering pandemic countermeasures more harmful than beneficial for the quality of medical care for indigenous populations [10,11]. With case fatality rates in the low single-digit percentage [12] and regional uncertainties regarding disease severity in indigenous populations [13,14], the weighing of the medical and socioeconomic consequences of the pandemic with population-based counterstrategies remained difficult and often highly emotional [15,16,17,18,19,20]. Notably, Australian authors recognized beneficial effects regarding the acceptance of anti-pandemic countermeasures if their choice and enforcement was controlled by the indigenous populations themselves, and thus not considered as externally imposed interventions [21].

The need to care for vulnerable indigenous populations had been already addressed in Columbia in the early stages of the pandemic [22], as it was soon recognized that indigenous communities with restricted health infrastructure were not only severely affected [23], but also particularly threated by the P.1 variant of concern (VOC) circulating in the Columbian Amazon basin [24]. In contrast, for indigenous populations such as the Wiwa, living in remote areas of the Sierra Nevada de Santa Marta in the north-east of Colombia, no information on the COVID-19 situation was available. The Wiwa—traditional agropastoralists—comprise a vulnerable group that usually avoids contact with the outside world. Although contacts between Wiwa and non-indigenous Columbians are rare, they do occur, especially in villages close to urban centres such as Valledupar or San Juan, and bear the risk of spreading SARS-CoV-2 within Wiwa populations. Further, contacts of Wiwa leaders with Colombian officials are potential sources of SARS-CoV-2 influx into indigenous communities. Poor socioeconomic conditions such as low levels of education and insufficient hygiene techniques, in combination with narrow and simple housing, may further facilitate the spread of SARS-CoV-2 in Wiwa communities. In addition, there is very restricted healthcare access. An average six-hour walk is needed to reach a hospital or health point, which makes it impossible for persons with severe COVID-19 to receive treatment. Even if health points are reached, the medical support is limited, as, e.g., oxygen, fluids, and necessary medication are not in place. Neither in the communities nor in the indigenous health points are SARS-CoV-2 tests available. Further, critical information on how SARS-CoV-2 is spread or how to prevent infection was not available in the rural areas during the study period. Hygiene measures, distancing and masks were largely unknown. Precise numbers about deaths and novel infections in this neglected population are not available; only verbal statements from the inhabitants on higher amounts of respiratory infections and higher death tolls might provide hints on pandemic activity. Even more, various health authorities doubted the existence of SARS-CoV-2 in indigenous communities.

We have chosen the village of San Juan because it is used by many Wiwas as a transit city, but the major part of the population are Colombians. To determine if COVID-19 had also reached Wiwa villages in the outback, we also selected villages located and listed in increasing distances from bigger cities such as Piñoncito, Sabana de Higuieron, Surimena and Ahuyamal.

By now, it has become possible to transfer patients and samples to the public health laboratory in Valledupar, which is equipped with a PCR laboratory and molecular diagnostic capacities, originally set up for a previous study on Chagas disease [25]. In addition, several months after the onset of the pandemic, some other laboratories began to offer PCRs, as well as other testing opportunities. However, the access for indigenous people is still scarce, as such tests are cost intensive and only available in larger cities, far apart from the living places of the Wiwas.

To shed light on SARS-CoV-2 manifestations and spread within these indigenous communities, COVID-19 screening using rapid diagnostic tests (RDTs) targeting SARS-CoV-2-antigen and SARS-CoV-2 antibodies, as well as SARS-CoV-2-specific real-time PCR, was conducted as a free-of-charge offer in areas predominantly inhabited by Wiwa populations as a cross-sectional assessment.

## 2. Materials and Methods

### 2.1. Study Population and Study Design

Between 28 March 2021 and 26 April 2021, a serology-, antigen- and PCR-based prevalence assessment for SARS-CoV-2 virus infections was conducted in north-eastern Colombian territories inhabited by the indigenous population of the Wiwas, as well as in San Juan as an urban centre. The individuals had not been vaccinated against SARS-CoV-2 previously and participated on a voluntary basis. Next to conducting the diagnostic assays as described below, clinical and epidemiological information on exact areas of origin, age, sex, ethnicity, blood pressure, heart rate, body temperature and blood oxygenation, as well as individually recorded clinical information with a potential relation to a previous or present COVID-19 disease, were collected.

A study team began to test for SARS-CoV-2 in various Wiwa villages. The number of inhabitants of these villages was used as a denominator to estimate the proportion of the population addressed. The testing was announced early enough to provide a sufficient time frame for the decision to participate in the study or not. If the volunteers agreed, one nasal swap was used for direct antigen testing; a second from the other nostril was used for real-time PCR in Valledupar. The presence of SARS-CoV-2-specific IgG and IgM antibodies was assessed using a serum sample collected for this purpose. All tests were performed independently of self-reported previous or present symptoms. In total, 452 persons were included in the study.

### 2.2. SARS-CoV-2 Diagnostic Tests

Active SARS-CoV-2 infection was detected by point-of-care antigen tests and bench-top real-time PCR from nasopharyngeal swabs (SteriCLIN, New stetic, S.A.) in viral transport medium (ad-bio Viral-U). For NAAT (nucleic acid amplification technique), the Allplex 2019-nCoV Real-Time PCR kit (Seegene, Seoul, Korea) with a limit of detection of 153.9 viral copies per mL sample and a specificity close to 100% [26] was used. The SARS-CoV-2 PCR-Kit was provided by den Instituto Nacional de Salud (INS), as this institution is in charge of distributing the kits for COVID-19 testing nationwide. Nucleic acid isolation from nasopharyngeal swaps was carried out with the MagaBio plus Virus DNA/RNA purification kit II (Luxus Lebenswelt GmbH, Willich, Germany) according to the manufacturer’s instructions. Antigen testing was performed using the rapid diagnostic test (RDT) assay Standard Q COVID-19 Ag Test (bestbion dx, Cologne, Germany). Evaluation against real-time PCR under standardized conditions suggested a sensitivity of 17.5% and a specificity close to 100% [27]. This antigen test was used as it was, in Colombia, the regionally most commonly used and distributed assay during the time period of the assessment.

SARS-CoV-2-serology was carried out using the RDT assay Bel TEST-It! SARS-CoV-2 IgM/IgG (PharmACT GmbH, Mannheim, Germany) which detects not only N-protein-directed antibodies, as the majority of POCT antibody tests do, but also spike-protein-specific and nucleocapsid-protein-specific IgG and IgM antibodies. The test manual stated sensitivity of 70.0% for IgM in the early stages of infection, of 92.3% for IgM in the late stages of infection and of 98.1% for IgG in the late stages of infection. Stages of infection were, however, not clearly defined by the manufacturer. As the specificity of SARS-CoV-2 antibody assays is known to vary considerably in different geographic regions [28], an in-house validation of specificity with 99 pre-pandemic regional serum samples taken from Wiwa volunteers [25] from the villages Tezhumake (n = 49) and Siminke (n = 50) was conducted prior to diagnostic use of the assay, which suggested a specificity close to 100%.

### 2.3. COVID-19 Case Definitions

Cases of active infections were defined as individuals who were positive for SARS-CoV-2 by antigen testing or PCR with or without positive serology. Previous infection was defined by being positive for SARS-CoV-2-specific IgG or IgM only.

### 2.4. Statistics

All data were analysed using SAS^®^, version 9.4 TS Level 1M5, or GraphPad Instat version 3.06 (GraphPad Software Inc., San Diego, CA, USA). For data description and unifactorial comparisons, univariable logistic regression including Wald’s test was used for both categorical and continuous data to compare SARS-CoV-2 negative with SARS-CoV-2 positive results. Three comparisons were introduced for active infections, for previous infections and for a combination of both, respectively. To ensure a multi-factorial perspective on the data and to control both for selection and possible confounding effects, a multivariable logistic regression was utilized for those factors which appeared statistically significant at the univariable level. Statistical significance was defined as *p* < 0.05.

### 2.5. Ethics

Ethical clearance was obtained from the Ethics Committee for Science of the University Area Andina, Valledupar, Cesar, Colombia (number 1304211). Prior to participation, written informed consent was provided by each participant. For minors, consent was given either by a parent or a legal guardian. All participants were informed about their results and received treatment, if appropriate. The study was performed in line with the declaration of Helsinki.

## 3. Results

### 3.1. Study Population and Basic Demographic Data

Between 28 March 2021 and 26 April 2021, a total of 452 individuals from 14 sites in north-eastern Colombia were assessed for present or previous contact with SARS-CoV-2. Their areas of origin, sorted in decreasing order based on number of participants, were: San Juan (n = 189), Sabana de Higuieron (n = 88), Piñoncito (n = 74), Ahuyamal (n = 44), Surimena (n = 33), Tezhumake (n = 11), Valledupar (n = 5), La Junta (n = 3), Atanquez (n = 1), Los Corazones (n = 1), Patillal (n = 1), Siminke (n = 1) and Vereda Platanal (n = 1). Table 1 indicates the proportion of screened inhabitants of those study sites.

The female:male ratio was 278:174, i.e., 61.5%:38.5%. The vast majority of 380 assessed individuals consisted of indigenous study participants; however, a minority of 72 non-indigenous individuals were also included in the assessment. Age information was available from 447 individuals, indicating a young study population with a median age of 29.2 years and an inter-quartile range from 17.5 to 42.1 years. The oldest participant was 82 years of age.

### 3.2. Molecular and Serological SARS-CoV-2 Diagnostic Test Results

The applied diagnostic algorithm recorded positive SARS-CoV-2 antigen tests in 22/452 (4.9%) instances and positive real-time PCR results in 61/452 (13.5%) cases, as well as 1/452 (0.2%) positive SARS-CoV-2-specific IgM assays and 125 (27.7%) positive SARS-CoV-2-specific IgG results. Based on the definitions above, the findings added up to 61/452 (13.5%) diagnosed cases of active infection and 113/452 (25.0%) diagnosed cases of previous infections. All positive antigen test results coincided with positive PCR results, resulting in a proportion of 36.1% active infections with sufficiently high viral load to allow positive antigen test results. In 13 out of 61 active infections (21.3%), already detectable anti-SARS-CoV-2-IgG suggested late stages of infection several weeks after the initial exposure or re-infections. In 4 of these 13 (30.8%) late-stage active infections or re-infections, positive antigen testing indicated high viral loads. Among the 113 patients with serologically proven previous infections, 112 (99.1%) patients were positive for IgG antibodies only and 1 patient (0.9%) for IgM antibodies only, respectively.

To increase visibility, the numbers and proportion of positive and negative PCR results, POCT antigen test results, POCT IgG results and POCT IgM results are depicted in Figure 1. Figure 2, in addition, shows combinations of positive test results.

### 3.3. Association of SARS-CoV-2 Test Results with Clinical and Epidemiological Outcomes

While our cross-sectional screening was ongoing, active infections were only recorded in Piñoncito (4/74, 5.4%) and San Juan (57/189, 30.2%), whereas previous infection based on serology results were seen in Ahuyamal (13/44, 29.5%), Piñoncito (23/74, 31.1%), Sabana de Higuieron, (10/88, 11.4%), San Juan (54/189, 28.6%), Surimena (1/33, 3.0%) and other, less intensively sampled sites (12/24, 50.0%). While female study participants showed an increased proportion of previous SARS-CoV-2-infections, regional indigenous populations were affected less by active SARS-CoV-2 infections than the tested non-indigenous Colombians. Regional differences in the distribution of SARS-CoV-2 activity were also seen (Table 2). When it came to the recorded clinical characteristics (Table 2), no major differences in clinical presentations were seen between individuals with active and previous infections compared to non-infected individuals. A slightly higher heart rate was found to be the only significantly different feature in actively infected COVID-19 patients (Table 2).

Taking into account that the enrolment of participants differed from village to village, logistic regression models to simultaneously adjust for differences in sex, ethnicity, village and heart rate were applied (Table 3), including all variables which showed statistical significance in the univariable model (Table 2). Here, it was not only shown that more males were classified as infected when active and previous infections were combined, but also that peripheral villages were overall even more severely affected by the pandemic than San Juan, where the highest proportion of active infections was measured in our study period.

As systematic interviews were not performed, infection-related symptoms were only based on verbal reports by the study participants. As shown in Table 4, the recorded symptoms were generally mild to moderate and did not allow us to further differentiate SARS-CoV-2-infections from other diseases. Of note, one male Wiwa from San Juan with active disease had reported a previously diagnosed SARS-CoV-2 infection half a year ago, and one female Wiwa from Piñoncito with active infection reported a positive rapid antigen test 20 days prior to the screening conducted here.

## 4. Discussion

To the best of our knowledge, our study provides, for the first time, information on SARS-CoV-2 dissemination in indigenous populations living in north-eastern Colombia. The results indicate seroprevalence estimates of SARS-CoV-2 antibodies of about 25%. As specificity assessments with serological SARS-CoV-2-antibody assays have shown varying diagnostic reliability depending on the geographic sample origin [28], an evaluation with pre-pandemic sera from Wiwa individuals [25] was carried out which suggested a high specificity.

Next to serological evidence of previous SARS-CoV-2 infections, we identified 13.5% active SARS-CoV-2 infections following antigen and real-time PCR testing within the study population. As expected for a virus infection spreading in outbreak-like waves, the spread was localized in individual regions. In this study, the highest activity of active infections per assessed individual was observed for San Juan, followed by Piñoncito. The regional SARS-CoV-2 spread suggests that—by chance—the assessment was performed while a local outbreak was ongoing. The finding of 30% active infections in assessed individuals from San Juan implies that local living conditions may facilitate SARS-CoV-2 transmission more easily than reported from over Colombian settings [22,23,24] in outbreak situations.

In contrast to previous evaluations, the proportion of positive antigen tests in relation to positive PCR tests was higher than expected [27], suggesting considerable viral loads in the respiratory secretions of the infected individuals. One Wiwa with PCR-confirmed SARS-CoV-2 infection reported previous COVID-19 half a year ago, and a second one a positive rapid antigen test three weeks prior to our assessment. Such hints in the medical history may indicate re-infections, either due to insufficient infection-induced immunity or emerging SARS-CoV-2 variants [24], or a prolonged shedding of SARS-CoV-2 in Wiwa individuals. In general, however, we did not record increased disease severity in participants with active infections, nor signs of relevant sequelae in convalescent individuals with detectable SARS-CoV-2-specific antibodies. The observed generally mild to moderate clinical COVID-19 courses could be (a) attributed to the young average age of our study population, (b) based on previous reports linking the poor living conditions of the indigenous to numerous infections [25,29,30] resulting in a well-regulated antiviral immune response, (c) attributed to the fact that severely ill patients would not have been able to walk for up to six hours to reach the closed screening facility, (d) based on the cultural educations which condemn complaining of mild to moderate symptoms and/or (e) based on the severe local stigmatization associated with SARS-CoV-2 infections, so symptoms are preferred to be concealed. Consequently, information on actual disease severity may not be free of bias.

Interestingly, from an epidemiological point of view, the proportion of non-indigenous study participants with active SARS-CoV-2 infection was higher compared to the indigenous population. Based on the concentration of active infections in San Juan, this higher proportion of active infections may support the hypothesis that SARS-CoV-2 could have been introduced into the indigenous communities from external sites and contacts. 

This is further strengthened by the multifactorial data analyses, which showed an extended risk for indigenous communities. In spite of the finding that nearly three times more previous SARS-CoV-2-infections were recorded in females (n = 83) than in males (n = 30) in the study population, the multifactorial approach suggested that contact with foreign communities by the male population facilitated transmission.

We observed a number of peculiarities in our assessment compared to the few other studies on COVID-19 in indigenous people in Colombia. First, a very recent nationwide cohort study reported a 1.2 hazard for COVID-19-associated mortality in the Colombian indigenous population compared to the non-indigenous Colombian population [31]. The lack of medical infrastructure in indigenous territories was blamed for the higher risk [23]. In contrast, we found a low mortality in the indigenous COVID-19 patients assessed in the present study. Probable reasons for this discrepancy have been summarized above. Second, in a previous study on COVID-19 in the Colombian indigenous population in the Amazon basin, a few SARS-CoV-2 introduction events into the local populations were sufficient to cause major outbreaks [24]. Due to the lack of molecular typing capacity in the present assessment, no comparable details on the local transmission chains within the Wiwa population can be provided.

The study has a number of limitations. We were only able to conduct a limited cross-sectional assessment on the impact of SARS-CoV-2 on indigenous Colombian populations within an arbitrarily chosen and limited time span. However, the primary intention of the study was to confirm that local indigenous populations were affected by COVID-19 infections in a quantitatively relevant dimension, a fact which was doubted by various local health officials, with the consequence of indigenous populations having scarce access to preventive measures, programs and vaccination schedules. The gained data therefore helped to improve the current situation, but also represent the most thorough assessment available from this region to our best knowledge. The limited numbers of 380 indigenous and 72 non-indigenous volunteers, of whom 72 non-indigenous volunteers were on average better educated and informed on COVID 19-disease than the Wiwas, can of course not be taken as completely representative for those populations. Additional studies are needed for further evaluation. As another limitation, the applied diagnostic SARS-CoV-2 PCR could not be freely chosen, as the decision of which kit to take was made by the INS.

Due to voluntary study participation, the ratio of males and females as well as the ethnicity ratio differed between villages and groups of COVID-19-negative and COVID-19-positive study participants. This causes crossing risk patterns in the strata defined by sex, communities and ethnicity, respectively, which makes an adjustment by multifactorial logistic regression crucial. However, the differences seen between uni- and multivariable results from these crossing patterns suggest that the enrolment of participants may introduce a residual confounding effect due to the different selection rates in the urban centre of San Juan and the Wiwa communities. While within the Wiwa villages a fairly good coverage suggests that no bias may be expected, no information on San Juan participants are available, which corresponds to a residual confounding effect. 

Last, due to budget constraints, neither sequence-based assessment of locally circulating SARS-CoV-2 variants nor a more detailed characterization of the adaptive immune response, such as evaluating the neutralization capacity of the humoral immune response or profiling the SARS-CoV-2 T-cell response with interferon-gamma release assays, were possible.

## 5. Conclusions

Our study provides a first insight into SARS-CoV-2 epidemiology and dissemination within indigenous Wiwa populations in north-eastern Colombia. Our results suggest previous SARS-CoV-2 infections in about 25% of the local population and a considerable disease activity, with about 13.5% exhibiting active infections at the time of the assessment, resulting in a proportion of 38.5% with confirmed contact with the virus. Our data do not only strongly argue to include the Wiwa population into surveillance and vaccination programs, but also to improve health policies to optimize health care for indigenous communities.

## Figures and Tables

**Figure 1 vaccines-09-01120-f001:**
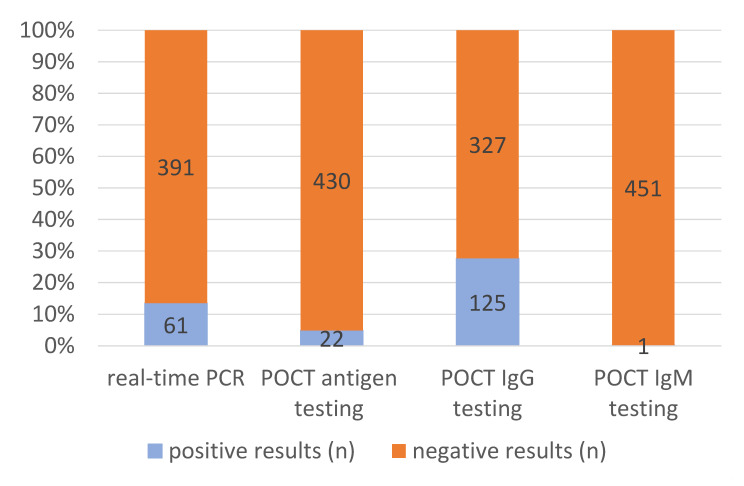
Numbers and proportions of positive test results.

**Figure 2 vaccines-09-01120-f002:**
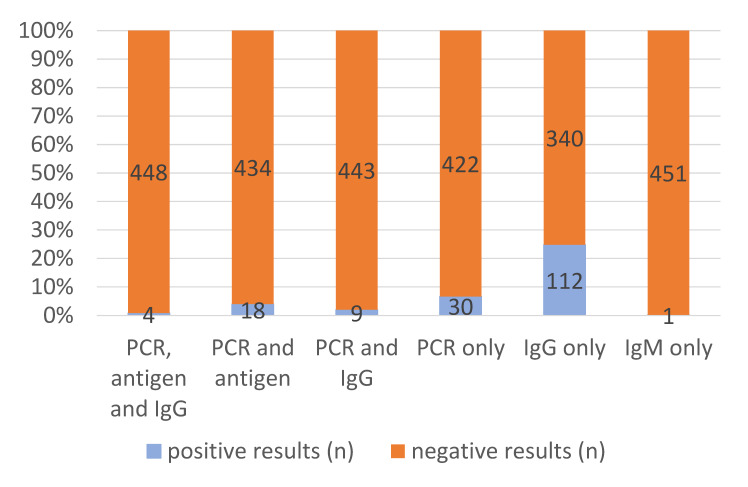
Numbers and proportions of positive results in various tests assays. Only combinations that really occurred are shown.

**Table 1 vaccines-09-01120-t001:** Proportion of assessed individuals per study site.

Origin of Study Participants	Number of Tested Individuals (n)	Estimated Number of Inhabitants (n)	Calculated Proportion of Screened Inhabitants (%)
San Juan	189	40,069	0.5%
Sabana de Higuieron	88	176	50.0%
Piñoncito	74	486	15.2%
Ahuyamal	44	109	40.4%
Surimena	33	106	31.1%
Others	24	534,697	<0.1%
Total	452	575,643	<0.1%

**Table 2 vaccines-09-01120-t002:** SARS-CoV-2 status by risk categories—results of univariable logistic regression analyses (all statistical tests within the logistic regression were compared with SARS-CoV-2 negative tests; for factors comparing groups the reference category is marked in bold).

**Risk Categories**	**Negative**		**Active**				**Previous**				**Active + Previous**			
	**n**	**%**	**n**	**%**	**OR**	** *p* **	**n**	**%**	**OR**	** *p* **	**n**	**%**	**OR**	** *p* **
Overall														
Total	278	61.5	61	13.5	x	x	113	25.0	x	x	174	38.5	x	x
Sex														
**Female (ref)**	156	56.1	40	14.4	1	x	82	29.5	1	x	122	43.9	1	x
Male	122	70.1	21	12.1	0.671	0.1773	31	17.8	0.483	0.0028	52	29.9	0.545	0.0031
Ethnicity														
**Indigenous (ref)**	243	64.0	47	12.4	1	x	90	23.7	1	x	137	36.1	1	x
Columbian	35	48.6	14	19.4	2.068	0.0402	23	31.9	1.775	0.0522	37	51,3	1.875	0.0152
Village														
Ahuyamal	31	70.5	0	0	x	0.9613	13	29.5	0.606	0.1811	13	29.5	0.295	0.0007
Piñoncito	47	63.5	4	5.4	0.116	<0.0001	23	31.1	0.707	0.2631	27	36.5	0.404	0.0014
Sabana de Higuieron	78	88.6	0	0	x	0.9386	10	11.4	0.185	<0.0001	10	11.4	0.090	<0.0001
**San Juan (ref)**	78	41.3	57	30.2	1	x	54	28.6	1	x	111	58.8	1	x
Surimena	32	97.0	0	0	x	0.9607	1	3.0	0.045	0.0027	1	3.0	0.022	0.0002
Other	12	50.0	0	0	x	0.9759	12	50.0	1.444	0.4086	12	50.0	0.703	0.4164
Age														
0−<15	62	65.3	14	14.7	0.912	0.8022	19	20.0	0.748	0.3630	33	34.7	0.810	0.4271
**15−<25**	48	56.5	9	10.6	0.757	0.5121	28	32.9	1.425	0.2365	37	43.5	1.173	0.5517
25−<45 (ref)	105	60.3	26	14.9	1	x	43	24.7	1	x	69	39.6	1	x
45−<60	38	67.9	7	12.5	0.744	0.5256	11	19.6	0.707	0.3707	18	31.1	0.721	0.3144
≥60	20	54.1	5	13.5	1.010	0.9860	12	32.4	1.465	0.3486	17	45.9	1.293	0.4802
Systolic blood pressure														
Systolic blood pressure per 1 mmHg	255	x	50	x	1.016	0.1299	106	x	1.007	0.3892	156	x	1.009	0.1693
Diastolic blood pressure														
Diastolic blood pressure per 1 mmHg	255	x	50	x	1.026	0.1232	105	x	0.977	0.0788	155	x	0.994	0.5900
Heart rate														
Heart rate per 1 beat per minute	278	x	61	x	1.032	0.0044	113	x	0.984	0.0937	174	x	1.002	0.7950
Body temperature														
Temperature per 1 °C	278	x	61	x	0.664	0.1289	113	x	0.748	0.1894	174	x	0.725	0.0836
O_2_ saturation														
O_2_ per 1%	277	x	61	x	1.058	0.5312	113	x	0.957	0.5239	174	x	0.990	0.8728

OR = odds ratio; *p* = *p*-value from Wald’s test within logistic regression; ref = reference; x = not applicable.

**Table 3 vaccines-09-01120-t003:** SARS-CoV-2 status by risk categories—results of multivariable logistic regression analyses (all statistical tests within the logistic regression were compared with the negative SARS-CoV-2 tests; for factors comparing groups the reference category is marked in bold).

**Risk Categories**	**Negative**		**Active**				**Previous**				**Active + Previous**			
	**n**	**%**	**n**	**%**	**OR**	** *p* **	**n**	**%**	**OR**	** *p* **	**n**	**%**	**OR**	** *p* **
Overall														
Total	278	61.5	61	13.5	x	x	113	25.0	x	x	174	38.5	x	x
Sex														
**Female (ref)**	156	56.1	40	14.4	1	x	82	29.5	1	x	122	43.9	1	x
Male	122	70.1	21	12.1	1.341	0.4050	31	17.8	2.033	0.0065	52	29.9	1.700	0.0203
Ethnicity														
**Indigene (ref)**	243	64.0	47	12.4	1	x	90	23.7	1	x	137	36.1	1	x
Columbian	35	48.6	14	19.4	2.394	0.0258	23	31.9	1.184	0.6240	37	51.3	1.476	0.1888
Village														
Ahuyamal	31	70.5	0	0	x	0.9608	13	29.5	1.932	0.1078	13	29.5	3.935	0.0004
Piñoncito	47	63.5	4	5.4	11.108	<0.0001	23	31.1	1.471	0.2557	27	36.5	2.884	0.0005
Sabana de Higuieron	78	88.6	0	0	x	0.9371	10	11.4	5.585	<0.0001	10	11.4	12.448	<0.0001
**San Juan (ref)**	78	41.3	57	30.2	1	x	54	28.6	1	x	111	58.8	1	x
Surimena	32	97.0	0	0	x	0.9599	1	3.0	23.873	0.0023	1	3.0	52.148	0.0001
Other	12	50.0	0	0	x	0.9752	12	50.0	0.872	0.7663	12	50.0	1.610	0.2873
Heart rate														
Heart rate per 1 beat per minute	278	x	61	x	0.984	0.2218	113	x	1.022	0.0459	174	x	1.007	0.4630

OR = adjusted odds ratio; *p* = *p*-value from Wald’s test within logistic regression; ref = reference; x = not applicable.

**Table 4 vaccines-09-01120-t004:** Recorded symptoms in screened individuals with active SARS-CoV-2 infection, with serological signs of previous infection and without any SARS-CoV-2 contact documented by laboratory results.

Recorded Symptom	Individuals with Active SARS-CoV-2 Infection (n = 61)	Individuals with Previous SARS-CoV-2 Infection (n = 113)	Individuals without Active or Previous SARS-CoV-2 Infection (n = 278)
Headache	3	2	2
Joint and muscular pain	1	-	2
Flu-like symptoms	1	2	-
Sore throat	-	2	2
Recent history of fever/chills	4	-	3
Loss of smell/taste	2	-	1
Fatigue	1	-	-
Rash	-	-	1

Only absolute numbers are presented. Statistical analysis was not performed as medical history had not been systematically recorded.

## Data Availability

All relevant data are presented in the manuscript. Raw data can be provided on reasonable request.

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
