# Peer review of "Direct and Indirect Proof of SARS-CoV-2 Infections in Indigenous Wiwa Communities in North-Eastern Colombia—A Cross-Sectional Assessment Providing Preliminary Surveillance Data"

_vaccines, 2021, doi:10.3390/vaccines9101120_

Round 1
Reviewer 1 Report
Reviewer #1:
Coronavirus disease 2019 (COVID-19) has emerged as a new world pandemic, infecting millions of people with a substantial mortality. There is significant interest study to analyzed the COVID-19 situation in indigenous communities.
Recently publications show several epidemiologist studies in poor countries and this review could be help in understand this pandemic.
In this manuscript, by Gustavo Concha et al titled "Direct and indirect proof of SARS-CoV-2 infections in indige- nous Wiwa communities in North-Eastern Colombia – a cross- sectional assessment providing preliminary surveillance data". The authors performed an analysis of serum samples from non-indigenous Colombians and determined IgG and IgM, antigen test and real-time PCR from nasal swabs.
There are several concerns that to be addressed.
This manuscript is well written and sites key findings in the field, therefore it will be helpful for epidemiological investigators entering into coronavirus/COVID-19 research. The study would benefit the section on information on SARS-CoV-2 dissemination in indigenous for the first time in North-Eastern Colombia.
Comments to improve the clarity of the manuscript are provided below.
Comments for the authors' consideration:
1.
- Double check small typing errors in all manuscript. For example:
- Line 212. SARS-CoV-2negative change to SARS-CoV-2 negative
- Line 223. SARS-CoV-2status change to SARS-CoV-2 status
- Please add a section with the figures results about the several techniques used antibody point-of-care test (POCT), antigen POCT test and real-time PCR.
Author Response
Coronavirus disease 2019 (COVID-19) has emerged as a new world pandemic, infecting millions of people with a substantial mortality. There is significant interest study to analyzed the COVID-19 situation in indigenous communities.
Recently publications show several epidemiologist studies in poor countries and this review could be help in understand this pandemic.
In this manuscript, by Gustavo Concha et al titled "Direct and indirect proof of SARS-CoV-2 infections in indige- nous Wiwa communities in North-Eastern Colombia – a cross- sectional assessment providing preliminary surveillance data". The authors performed an analysis of serum samples from non-indigenous Colombians and determined IgG and IgM, antigen test and real-time PCR from nasal swabs.
There are several concerns that to be addressed.
This manuscript is well written and sites key findings in the field, therefore it will be helpful for epidemiological investigators entering into coronavirus/COVID-19 research. The study would benefit the section on information on SARS-CoV-2 dissemination in indigenous for the first time in North-Eastern Colombia.
Comments to improve the clarity of the manuscript are provided below.
Comments for the authors' consideration:
1.
- Double check small typing errors in all manuscript. For example:
- Line 212. SARS-CoV-2negative change to SARS-CoV-2 negative
- Line 223. SARS-CoV-2status change to SARS-CoV-2 status
Thank you for your hint. We corrected those mentioned above, but also others.
- Please add a section with the figures results about the several techniques used antibody point-of-care test (POCT), antigen POCT test and real-time PCR.
As requested, respective figures (new figures 1 and 2) have been added to visualize the differentiated pattern of positive test results (Results chapter, sub-heading 3.2. “Molecular and serological SARS-CoV-2 diagnostic test results”).
Reviewer 2 Report
The topic of "Direct and indirect proof of SARS-CoV-2 infections in indigenous Wiwa communities in North-Eastern Colombia – a cross-sectional assessment providing preliminary surveillance data" is interesting and meaningful. It focused on SARS-CoV-2 epidemiology and spread in indigenous communities in North-Eastern Colombia, which provide an efficient and essential evidence for policy-makers on screening and vaccination programs. However, the following questions should be answered:
1. How can you proved the sample size is enough and efficient in the study? Which should be clearly described?
2. There were 380 indigenous and 72 non-indigenous volunteers included in you study. Does the proportion could reflect the SARS-CoV-2 infectious real situation between indigenous and non-indigenous?
3. The time span of this study was very limited. More persuasive and reasonable results could be get if you can get sufficient funding.
4. The reason why these methods of SARS-CoV-2 diagnostic tests were chose should be further described.
Author Response
The topic of "Direct and indirect proof of SARS-CoV-2 infections in indigenous Wiwa communities in North-Eastern Colombia – a cross-sectional assessment providing preliminary surveillance data" is interesting and meaningful. It focused on SARS-CoV-2 epidemiology and spread in indigenous communities in North-Eastern Colombia, which provide an efficient and essential evidence for policy-makers on screening and vaccination programs. However, the following questions should be answered:
- How can you proved the sample size is enough and efficient in the study? Which should be clearly described?
We did not perform a formal sample size calculation or randomization, since this was an observational study based on voluntary participants, as outlined in material and methods. The main aim of the study was, to give proof that the indigenous population was affected by COVID-19, as this was doubted by various health officials. The sample size gained could show the impact of COVID-19 in the Wiwas, which was demonstrated to the health officials and gave the indigenous populations access to the programs. We have summarized this in the introduction (line 88,89) and discussion part (please see also answer to point 2).
- There were 380 indigenous and 72 non-indigenous volunteers included in you study. Does the proportion could reflect the SARS-CoV-2 infectious real situation between indigenous and non-indigenous?
We have described this in a new paragraph in the discussion section: “We were only able to conduct a limited cross-sectional assessment on the impact of SARS-CoV-2 on indigenous Colombian populations within an arbitrarily chosen and limited time span. However, the primary intention of the study was to confirm that local indigenous populations were affected by COVID-19 infections in a quantitatively relevant dimension, a fact which was doubted by various local health officials with the consequence of indigenous populations having scarce access to preventive measures, programs and vaccination schedules. The gained data therefore helped to improve the current situation, but also represents the most thorough assessment available from this region to our best knowledge. The limited numbers of 380 indigenous and 72 non-indigenous volunteers, of whose 72 non-indigenous volunteers were on average better educated and informed on COVID 19-disease than the Wiwas, can of course not be taken as completely representative for those populations. Additional studies are needed for a further evaluation“.
- The time span of this study was very limited. More persuasive and reasonable results could be get if you can get sufficient funding.
Yes, we do agree completely. However, with this study we wanted to solve an acute problem: we wanted to change the opinion of health officials, showing, that COIVD-19 is present in indigenous populations. Therefore, the study was urgently initiated to improve the situation as soon as possible. Consequently, funding and time were limited, however, the shown data still represent the most thorough assessment available from this region to our best knowledge. We implemented this also in the new paragraph (see also answer 2).
- The reason why these methods of SARS-CoV-2 diagnostic tests were chose should be further described.
The PCR test was advised by the Instituto Nacional de Salud (INS), who is a Colombian institution that distributes nationwide COVID-PCR tests to their reference laboratories. Our former Chagas PCR laboratory in Valledupar, which is a fully equipped PCR laboratory, was upgraded to a Corona Reference Laboratory, therefore we used this laboratory, going along with the INS provided kits. In addition, it was important for us to perform all tests in an official reference laboratory, so that there could not be any doubt about the found results.
We have chosen a rapid point of care antibody test to be able to directly communicate results to the participants. The test chosen was the one, that was most commonly used in that area.
In addition, we selected that particular point of care test from PHARM ACT, as it does not only detect N-protein directed antibodies, as the majority of POCT antibody tests do, but also spike protein-specific and nucleocapsid protein-specific IgG and IgM antibodies, which is advantageous compared to most other available POCT in Colombia.
We have added the information in the method and material part: ”The SARS-CoV-2 PCR-Kit was provided by den Instituto Nacional de Salud (INS), as this institution is in charge to distribute the kits for Corona-testing nationwide” and “This antigen test was used as it was in Colombia the regionally most commonly used and distributed assay during the time period of the assessment” and “SARS-CoV-2-serology was carried out using the RDT assay Bel TEST-It! SARS-CoV-2 IgM/IgG (PharmACT GmbH, Mannheim, Germany) which detects not only N-protein directed antibodies, as the majority of POCT antibody tests do, but also spike protein-specific and nucleocapsid protein-specific IgG and IgM antibodies”.
We have also added this point to the discussion: “ As another limitation, the applied diagnostic SARS-CoV-2 PCR could not be freely chosen, as the decision which kit to take was made by the INS”.